# Improving Access to Radiotherapy Services in Gauteng: Quantitative Analysis of Key Time Intervals from Diagnosis to Treatment

**DOI:** 10.3390/ijerph22040544

**Published:** 2025-04-02

**Authors:** Portia N. Ramashia, Pauline B. Nkosi, Thokozani P. Mbonane

**Affiliations:** 1Department of Medical Imaging & Radiation Sciences, Faculty of Health Sciences, University of Johannesburg, Johannesburg 2000, South Africa; 2Faculty of Health Sciences, Durban University of Technology, Durban 4000, South Africa; paulinen1@dut.ac.za; 3Department of Environmental Health, Faculty of Health Sciences, University of Johannesburg, Johannesburg 2000, South Africa; tmbonane@uj.ac.za

**Keywords:** access to care, cancer care continuum, healthcare system, public health, radiotherapy, South Africa, time intervals, treatment delay, treatment planning, waiting times

## Abstract

Background: Timely access to radiotherapy is crucial for optimal cancer treatment outcomes, as delays in receiving treatment can lead to disease progression and decreased chances of survival. Healthcare systems need to prioritise efficient scheduling and coordination of radiotherapy services to ensure patients receive timely care. This study aims to quantitatively analyse the key time intervals in the cancer care continuum, specifically from diagnosis to the start of radiotherapy treatment in Gauteng. Methods: Data from 800 patients treated between January and December 2023 were analysed using a retrospective analysis of patient records from the two public radiotherapy centres in Gauteng Province, one in Johannesburg and the other in Pretoria, focusing on key time intervals in the cancer care continuum. The durations from diagnosis to the commencement of radiotherapy were analysed. Results: The mean duration of the first consultation was 8.32 months, highlighting significant delays in accessing specialised care. This finding is further supported by the average time until a Computed Tomography (CT) simulation, which was 13.38 months, highlighting a considerable delay in treatment planning. Conclusion: This study highlights systemic delays in the Gauteng radiotherapy pathway, highlighting the need for optimisation of referral processes, resource constraints, and strategies to improve cancer care.

## 1. Introduction

Cancer is a significant public health issue worldwide, and radiotherapy is an essential treatment modality for many cancer patients [1]. However, access to radiotherapy services in low- and middle-income countries is limited, which leads to significant disparities in cancer outcomes. South Africa is no exception to this problem, and the province of Gauteng faces a rising cancer burden coupled with challenges in ensuring timely access to radiotherapy services [2,3,4]. Timely access to radiotherapy is critical for optimal cancer treatment outcomes. Delays in access can negatively impact patient outcomes and survival [5]. Several factors contribute to these delays in radiotherapy access. For example, socioeconomic challenges faced by patients can significantly impact their ability to seek and receive timely treatment. To address these challenges, this study is part of a larger, multi-phase research project aimed at developing a comprehensive framework for improving access to radiotherapy services in Gauteng. This larger project encompasses several phases, including a descriptive cross-sectional study on patients’ experiences of socioeconomic and demographic challenges to radiotherapy access, and an exploration of structural quality indicators for radiotherapy. The specific aim of this study was to quantitatively analyse key time intervals in the radiotherapy care pathway, from diagnosis to treatment initiation, to identify potential bottlenecks and areas for improvement. By quantifying these time intervals and identifying potential delays, this study provides crucial data for informing the development of targeted interventions and strategies to improve radiotherapy access in Gauteng, contributing to the overarching goal of the larger research project.

Access to radiotherapy in low- and middle-income countries (LMICs) like South Africa is troubled with challenges, delaying timely and equitable treatment for many patients [6]. Many LMICs have a critical shortage of radiotherapy centres, particularly in rural and underserved areas. This geographic disparity forces patients to travel long distances, often at significant personal cost, to access treatment. There is a severe shortage of qualified radiation oncologists, medical physicists, radiation therapists, and oncology nurses in LMICs [7,8,9,10,11]. This shortage limits the capacity to deliver timely and high-quality radiotherapy services. Low levels of public awareness about cancer and the benefits of radiotherapy can lead to late presentations and delayed treatment seeking [12]. In some cases, primary care providers may have limited knowledge about radiotherapy referral pathways, further contributing to delays in accessing treatment [13].

Timely access to radiotherapy is not just about providing treatment, it is about improving survival rates, enhancing the quality of life, and ensuring equitable access to life-saving cancer care [14,15,16]. Radiotherapy is most effective when delivered in a timely manner after diagnosis [6]. Delays allow cancer cells to proliferate and potentially spread, making treatment more challenging and potentially less effective [5,6,17]. The cancer care continuum encompasses the entire patient journey, from the initial suspicion of cancer to treatment, survivorship, and beyond. The continuum includes the time from a patient first experiencing symptoms to seeking medical attention. This interval is influenced by factors like symptom awareness, perceived seriousness, and barriers to healthcare access. Then, the time from the first medical consultation to a confirmed cancer diagnosis, which involves various tests and procedures, and delays can occur due to factors like test availability, specialist referrals, and administrative processes [18,19,20].

The focus of this phase of the research is the time from diagnosis (confirmed biopsy results) to the treatment initiation. The phase involves reviewing patients’ files for the timelines from the biopsy date to the start of radiation. Analysing these intervals helps identify bottlenecks in the healthcare system.

This paper presents findings from a phase of a broader postgraduate research project aimed at developing a practical, evidence-based strategic framework for improving radiotherapy access for cancer patients in Gauteng. This phase of the project aimed to quantitatively analyse the key time intervals within the cancer care continuum, from the point of diagnosis to the initiation of radiotherapy treatment in Gauteng. By examining these intervals, the study provides valuable insights into the accessibility of radiotherapy services in the province. Moreover, the insights shed light on potential components of a framework that will improve timely and equitable access to this life-saving treatment modality.

## 2. Materials and Methods

### 2.1. Study Design and Site

A quantitative descriptive retrospective cross-sectional study design was used to achieve the study objectives. The study was conducted in public hospitals located in the City of Johannesburg and Tshwane within the Gauteng Province, South Africa.

### 2.2. Study Participants

This retrospective study included all adult patients (18 years or older) diagnosed with cancer and treated with external beam radiotherapy at the two public radiotherapy facilities in Gauteng Province between January and December 2023 for the five most commonly diagnosed cancers (breast, head and neck, prostate, gastrointestinal, and cervical).

Inclusion criteria: Patients were included if they fit the following criteria:They were 18 years of age or older;They had a histologically confirmed diagnosis of cancer;They received external beam radiotherapy as part of their treatment;They had complete data available for the key time intervals of interest (date of diagnosis, date of radiation oncology consultation, date of CT simulation, and date of start of radiotherapy).

Recruitment Method: Patient data were retrospectively extracted from the cancer the treatment planning lists of the two public radiotherapy facilities in Gauteng Province. All files meeting the inclusion criteria were included. No direct patient contact or active recruitment was involved in this study.

### 2.3. Data Collection

Data were collected by retrospectively reviewing patient files from January to December 2023 to extract data on key time intervals. The 2023 data were collected to ensure that the data for the study were current and relevant. Data extraction was performed by the primary researcher and a qualified research assistant using a standardised data collection sheet adapted from the survey by the IAEA that used validated quality indicators. To ensure comprehensive data capture, the researchers divided the patient files, with each being responsible for extracting data from different sections. To ensure accuracy, each researcher then independently reviewed the data extracted by the other, serving as a second check. Furthermore, an independent statistician had access to the raw data and was consulted to verify data integrity and resolve any uncertainties or discrepancies.

The retrospective nature of this study meant that missing data were a potential concern. Not all patient files contained complete information for all the key time intervals under investigation. Files with incomplete data for any of the key time intervals were excluded from the analysis and replaced with the next file in the list that met the criteria. The study included files of cancer patients who received radiotherapy for the five most diagnosed cancers in South Africa. A sample size of 800 patient files (400 from each facility) was chosen to provide adequate statistical power to detect clinically meaningful differences in the key time intervals of the radiotherapy care pathway and to allow for subgroup analyses by cancer type and facility. With each facility treating approximately 4000 patients annually, the selection represents a 10% sample of the total patient population at each centre.

A systematic approach was used, where radiotherapy planning records were reviewed by including the first 33 to 34 patient files per month that met the inclusion criteria for the five most common cancers (breast, head and neck, prostate, gastrointestinal, and cervical), including 34 files for January, May, September, and October, and 33 files for all other months. To ensure representation across the year and to account for potential seasonal variations in patient load or cancer type, the sampling was conducted monthly. Within each month, the first 33–34 files meeting the inclusion criteria were selected. There was no additional randomisation within this interval. While a fully randomised selection would be ideal, the systematic approach was chosen for its feasibility in the context of a retrospective chart review. Given the large sample size and the lack of any known cyclical patterns in patient characteristics or file ordering, this systematic approach provides a reasonable approximation of a random sample.

The data collection sheet (Appendix A) employed was adapted from the survey by the IAEA that used validated quality indicators. The following time intervals were included in the analysis:Time from diagnosis to first consultation with a radiation oncologist;Time from diagnosis to CT simulation for radiotherapy planning;Time from CT simulation for radiotherapy planning to initiation of radiotherapy.

### 2.4. Study Key Terms

The study was conceptualised using Andersen Newman’s behavioural model of health service utilisation. This model suggests that health service utilisation is determined by three main components: predisposing characteristics (e.g., demographics, social structure, health beliefs), enabling resources (e.g., income, insurance, access to care), and need (perceived and evaluated). These components influence an individual’s decision to seek and utilise health services. Therefore, for this study, the term ’access’ is defined as the means through which patients enter the health system and continue with the treatment process, in this case, radiotherapy [21]. The term ‘key time interval’ represents the distinct periods within the cancer care continuum, explicitly focusing on the timeframe between a patient’s initial cancer diagnosis and the commencement of radiotherapy treatment in Gauteng. These intervals highlight potential areas where delays may occur, impacting the timely delivery of radiotherapy services [22].

### 2.5. Data Management and Analysis

Data were entered into Microsoft Excel and then analysed using the IBM SPSS Version 29 software. Descriptive statistics were used to summarise patient characteristics and time intervals. Categorical variables were summarised with counts and percentages for the key time intervals; median, minimum, and maximum values were used. A *t*-test was conducted to identify whether there were significant differences in waiting time between the two hospitals and analysis of variance. To compare the different cancers, because there are five primary cancers, the one-way ANOVA test with the Brown–Forsythe test was used. The Pearson chi-squared tests were conducted to identify significant differences between the hospitals in terms of the types of cancers [23].

A Bonferroni Adjustment was applied, where the significance level was divided by five and 0.05 by five, and the new significance level was 0.01 for the multiple comparisons for the five different cancers analysis. The *p*-values below the critical value of 0.01 were interpreted as statistically significant [23].

### 2.6. Ethical Considerations

Ethical approval for this study was obtained from the University of Johannesburg (registered as HDC-01-154-2023 and REC-2509-2023), the Johannesburg Health District, and the City of Johannesburg (registered as NHRD ref no.: GP_202311_078). Patient confidentiality was rigorously maintained throughout the study. Identifiable information was anonymised, and the data were stored securely.

## 3. Results

### 3.1. Sample Characteristics

Table 1 below summarises the key demographic data of the study sample.

The study comprised 71.4% (n = 571) female patients and 28.6% (n = 229) male patients. The age group 46 to 55 years old had the highest representation in the study at 28% (n = 230), followed by the age group 56 to 66 years old at 24.8% (n = 198). Regarding the cancer types, 46.5% of the files were related to cervical cancer and 18.3% were related to breast cancer. Furthermore, in terms of the stage at diagnosis reported in the patient files, most patients were at stage 3 (47.9%) and stage 2 (35.9%). Table 2 and Figure 1 below show the distribution of the diagnosis and stage at diagnosis in the two centres.

The referral pattern for the radiation centre in Johannesburg shows that the highest number of patients, 41%, were referred from the Chris Hani Baragwanath Academic Hospital. The internally referred patients from the hospital with the radiation facility accounted for 19%. In the radiation centre in Pretoria, most of the patients were referred from Dr George Mukhari Hospital, with 27% of the patients being referred internally. Moreover, this centre in Pretoria reported that 24% of the patients were referred from outside the province, including 8.8% (n = 70) from Mpumalanga, 3.3% (n = 26) from North West, and 0.2% (n = 2) from Limpopo.

The treatment outline of the files collected was as follows: Chemotherapy was administered to 49.4% of patients in the cohort, either before or after surgery and radiation therapy. There were 29% of patients who had undergone surgery before or after chemotherapy and before radiation therapy. The decision to not consider surgery for 71% of patients may also be linked to their stage at diagnosis. Furthermore, sadly, included in the collected data, 3.9% (n = 31) of patients were reported as having passed away since they started visiting the Radiation Oncology department between simulation and starting treatment. Regarding the radiotherapy prescriptions, most patients, totalling 87%, underwent treatment with radical (curative) intent.

Furthermore, the data encompassed patients with approved treatment plans who did not initiate radiotherapy. Out of the 50 files (6.3%), the reasons for not starting radiotherapy included patients who defaulted (n = 27), patients who passed away (n = 22), and one patient who was transferred to another hospital. Moreover, there were 25 patients whose treatment plans were changed during or before the start of radiotherapy, accounting for 3.1% of the files due to reasons such as progressive disease (n = 17), poor response (n = 3), and treatment stopped (n = 5) for various reasons.

### 3.2. Key Time Intervals

Understanding the time intervals within the cancer care continuum, from diagnosis to the initiation of radiotherapy treatment, is crucial for identifying potential bottlenecks and areas for improvement in access to timely care. The key time intervals provide insights into the patient journey and highlight potential areas where interventions could be implemented to improve access to radiotherapy services. These intervals, as shown in Table 3, include the time from diagnosis (biopsy date) to the first consultation with a radiation oncologist, the time from diagnosis (biopsy date) to CT simulation for radiotherapy planning, and the time from CT simulation for radiotherapy planning to initiation of radiotherapy. Analysing these intervals will help to pinpoint delays and inform strategies to optimise the delivery of radiotherapy services.

#### 3.2.1. Time from Diagnosis to First Consultation with a Radiation Oncologist

The duration between the cancer diagnosis date (biopsy date) and the first radiation oncology presentation date is a key time interval that can impact patient outcomes. The mean duration between the cancer diagnosis date and the first radiation oncology presentation (Table 4) was 8.32 months (SD = 10.906, min = −39.33, max = 1114.33). This indicates that, on average, patients experienced a delay of 8 months between their cancer diagnosis date and their first consultation with the radiation oncologist. The minimum value of −39.33 months is an anomaly that was found in one file. Upon further investigation of the original record, we found that the patient was referred to the radiation oncology department following initial imaging. The definitive biopsy diagnosis was confirmed later. This unusual sequence highlights the importance of considering the clinical context in interpreting these time intervals.

The mean duration between the first radiation oncology presentation and cancer diagnosis at the centre in Johannesburg was 7.17 months (SD = 11.6, min = −39.33, max = 85.78), and the mean duration at the centre in Pretoria (Figure 2) was 9.49 months (SD = 10.04, min = 0.39, max = 114.33). There was no significant difference in the mean duration between the centre in Johannesburg and the centre in Pretoria (*p*-value = 0.003).

#### 3.2.2. Time from Diagnosis to CT Simulation for Radiotherapy Planning

The mean duration between the diagnosis date and radiotherapy CT simulation (Table 3) was 13.38 months (SD = 17.25, min = 0.00, max = 116.04). This finding reveals that following diagnosis, patients waited an average of 13 months before undergoing the CT simulation process, which is a critical step in treatment planning for radiotherapy. When comparing the two radiation centres, there was no significant difference between the two centres (*p*-value = 0.002). The mean durations (Figure 3 and Figure 4 were 15.26 and 11.51 (SD = 21.89 and 10.46; min = 0.00 and 0.99; max = 112.23 and 116.04) for the radiation centres in Johannesburg and Pretoria, respectively. This delay in undergoing radiotherapy CT simulation may have implications for the overall treatment timeline and effectiveness of the therapy.

#### 3.2.3. Time from CT Simulation for Radiotherapy Planning to Initiation of Radiotherapy

The mean duration between radiotherapy CT simulation and the start of radiotherapy treatment (Table 3) was 9.23 weeks (SD = 6.33, min = 0.00, max = 63.14). This finding demonstrates that, on average, there were 9.3 weeks between CT simulation and the actual initiation of radiotherapy. The mean duration between radiotherapy CT simulation and the start of radiotherapy treatment (Figure 5) at the centre in Johannesburg was 11.43 weeks (SD = 7.84, min = 0, max = 63.14) and for the centre in Pretoria, the mean duration was 7.30 weeks (SD = 3.69, min = 0.43, max = 24.14). The difference in mean duration between the two centres was statistically significant (*p*-value = 0.00).

#### 3.2.4. The Key Time Intervals Stratified by Cancers

The mean duration between cancer diagnosis and the first radiation oncology presentation was not significantly different between all the cancers except for breast cancer and prostate cancer (*p*-value = 0.005). Prostate and breast cancer had a significantly longer mean duration than all cancers (Figure 6), 19.09 and 12.85 months, respectively. The lowest mean duration was seen in head and neck cancer, at 3.71 months. The *p*-value of 0.005 indicates that the difference in mean duration between the cancers is statistically significant.

Additionally, the mean duration (in months) between the diagnosis date and radiotherapy CT simulation showed the same pattern as the mean duration between cancer diagnosis and first radiation oncology presentation. With the highest mean duration of 38 and 18.21 months for prostate and breast cancer patients, respectively (Figure 7). The mean duration for cervical, gastrointestinal, and head and neck cancers was around 5 months (5.77, 5.27, 5.24). Patients with cervical cancer had the shortest mean time in weeks (Figure 8) between the start of radiotherapy treatment and the radiotherapy CT simulation, at 6.73 weeks, and patients with head and neck cancer had the longest, at 6.8 weeks. The study also found that breast cancer patients had the most prolonged mean duration between radiotherapy CT simulation and the start of radiotherapy treatment at 8.42 weeks. These findings suggest variations in the time intervals between diagnosis, treatment planning, and the initiation of radiotherapy among different cancer types.

## 4. Discussion

This study aimed to identify key bottlenecks in the radiotherapy care pathway for cancer patients in Gauteng Province, South Africa, revealing significant variations in treatment duration across cancer types and centres. However, it is important to consider these findings when considering the study’s limitations. As a retrospective analysis, this research is subject to the inherent constraints of relying on pre-existing data.

The results presented in this study highlight significant delays in the radiotherapy care pathway for cancer patients in Gauteng, SA. The average delay of 8.32 months between diagnosis and first radiation oncology consultation is concerning and could be attributed to various factors, including delays in referral, scheduling issues, limited access to specialists, or patient-related factors. These results indicate that, on average, patients experienced a delay of 8 months between their cancer diagnosis date and their first consultation with the radiation oncologist. Delays in diagnosis and treatment can lead to disease progression and a poorer prognosis for cancer patients. Therefore, reducing the time between the consultation with the radiation oncologist and the cancer diagnosis date is crucial for improving outcomes and survival rates [20,24,25].

The average 13.38-month wait for radiotherapy CT simulation after diagnosis is a substantial delay that could impact the effectiveness of treatment. The results also reveal significant variations in the mean duration between the Johannesburg and Pretoria centres. The results indicate that centre-specific factors, such as resource availability, staffing levels, or operational efficiency, may contribute to these discrepancies [13].

There are concerns raised by the fact that people with prostate and breast cancer have much longer durations than people with other types of cancer. The longer durations may be because they need neoadjuvant therapy. Further investigation is needed to understand the underlying reasons for these differences, as they may point to unique barriers or challenges faced by patients with these specific cancer types [16,19,20].

The results of this study are consistent with previous research that has identified delays in radiotherapy as a significant barrier to optimal cancer treatment outcomes [26,27,28]. These delays can result in disease progression and decreased survival rates for patients, highlighting the importance of addressing this issue in clinical practice [29,30]. Previous research has shown that patient delay, defined as the time between symptom discovery and first presentation to a healthcare facility of more than 3 months, makes up most of the total delay time. Factors associated with late presentation include low education, lack of knowledge about cancer symptoms, and poor emotional and physical well-being [13]. Disparities in timeliness of care have also been reported, with studies finding that insurance status, geographic region, and type of treatment hospital are also associated with treatment delays [31,32].

## 5. Strengths and Limitations

This study provides valuable insights into the radiotherapy care continuum in Gauteng, South Africa, but it is essential to acknowledge both its strengths and limitations. The strengths of the study were that it was conducted using real-world data from only two cancer centres, enhancing the generalizability of the findings to similar settings. By examining radiotherapy access, the study addresses a crucial aspect of cancer care that directly impacts patient outcomes. Moreover, the study analysed a large dataset, allowing for a detailed examination of time intervals. As a retrospective study, it relies on existing data, which may be subject to limitations such as missing information or inconsistencies in data collection methods. This phase of the study primarily relied on quantitative data, which may not fully capture the nuances of patient experiences, referral processes, or institutional barriers; however, these were explored in the other phases of the research project that are not reported in this paper. The other phases included a descriptive cross-sectional study on patients’ experiences of socioeconomic and demographic challenges to radiotherapy access and an exploration of structural quality indicators for radiotherapy to understand better the patient experiences and the institutional factors driving the delays.

## 6. Conclusions

This study aimed to analyse the key time intervals in the radiotherapy care pathway for cancer patients in Gauteng Province, South Africa, identifying significant delays and variations across cancer types and treatment centres. The findings reveal critical bottlenecks that require targeted interventions to improve access to radiotherapy services. Specifically, the diagnosis-to-consultation interval was found to be exceptionally long, averaging 8.32 months. Based on this finding, we recommend prioritising interventions to streamline the referral process and reduce specialist appointment wait times. The interventions could involve implementing standardised referral protocols across all healthcare facilities in Gauteng, as well as increasing the availability of radiation oncologists through strategic recruitment and resource allocation incorporating Artificial Intelligent (AI)-powered tools.

Furthermore, the significant variations observed in the mean duration between the Johannesburg and Pretoria centres suggest the need for a thorough assessment of centre-specific resource allocation, staffing levels, and operational efficiency. Addressing these disparities will require a commitment to equitable resource distribution and the implementation of best-practice protocols across all radiotherapy centres in the province. Finally, the longer durations observed for prostate and breast cancer patients, potentially due to neoadjuvant therapy, warrant further investigation. Future research should focus on optimising the coordination of neoadjuvant therapy and radiotherapy to minimise delays and improve treatment outcomes for these specific patient populations.

In conclusion, this study provides a valuable baseline for monitoring the impact of future interventions aimed at improving access to radiotherapy services in Gauteng. By focusing on the concrete, prioritised interventions outlined above, including the strategic implementation of AI technologies, healthcare policymakers and administrators can take meaningful steps towards reducing delays and ensuring equitable access to timely and effective cancer treatment for all patients.

## Figures and Tables

**Figure 1 ijerph-22-00544-f001:**
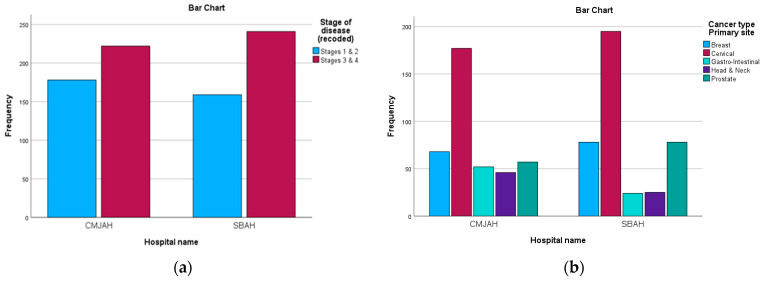
(**a**) describes the distribution of cancer stages; (**b**) describes the distribution of cancer types.

**Figure 2 ijerph-22-00544-f002:**
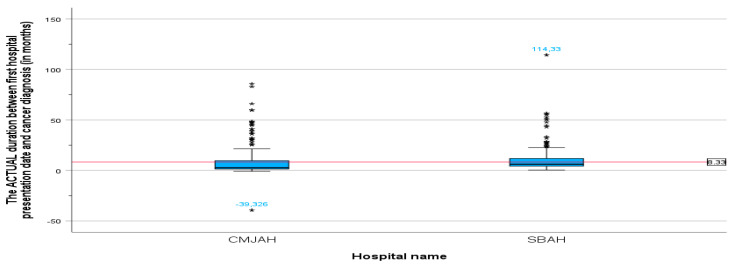
Duration between cancer diagnosis and first radiation oncology consultation in months. * Indicates the cases in the actual duration.

**Figure 3 ijerph-22-00544-f003:**
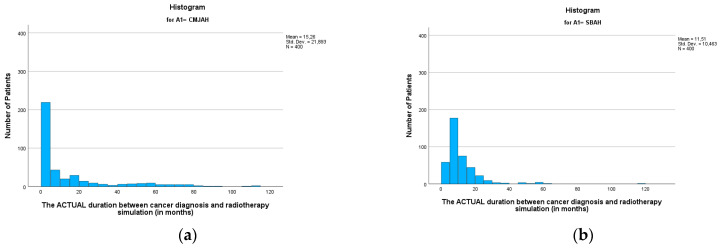
Histogram showing duration between diagnosis and CT simulation in (**a**) Johannesburg and (**b**) Pretoria (*p*-value = 0.002).

**Figure 4 ijerph-22-00544-f004:**
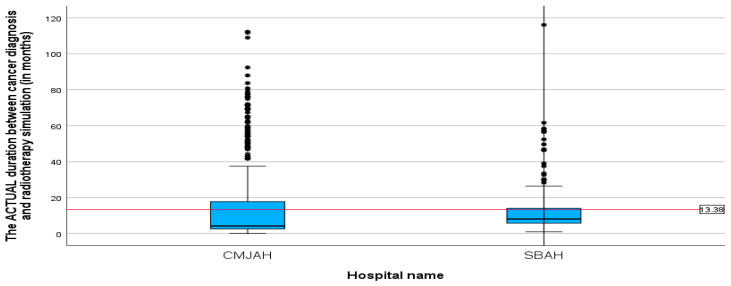
Boxplots showing duration between cancer diagnosis and CT simulation in months (*p*-value = 0.002). * Indicates the cases in the actual duration.

**Figure 5 ijerph-22-00544-f005:**
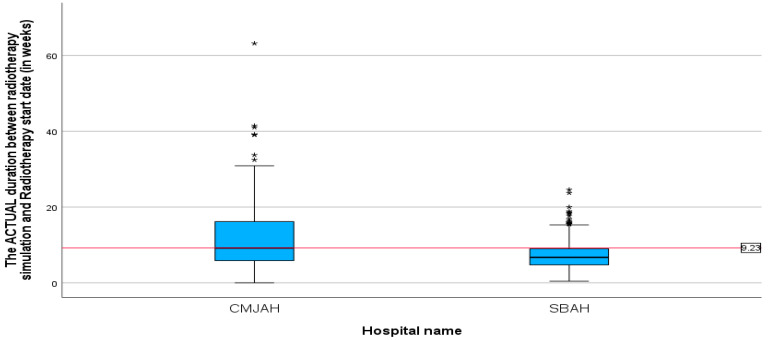
Boxplots showing the duration between CT simulation and the start of radiotherapy in weeks (*p*-value = 0.00). * Indicates the cases in the actual duration.

**Figure 6 ijerph-22-00544-f006:**
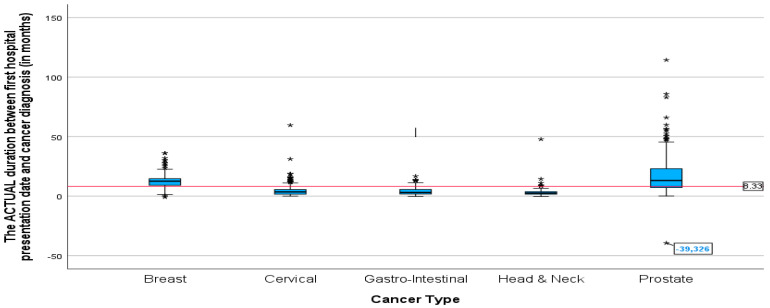
Boxplots showing duration between cancer diagnosis and first radiation oncology consultation for different cancers in months (*p*-value = 0.005). * Indicates the cases in the actual duration.

**Figure 7 ijerph-22-00544-f007:**
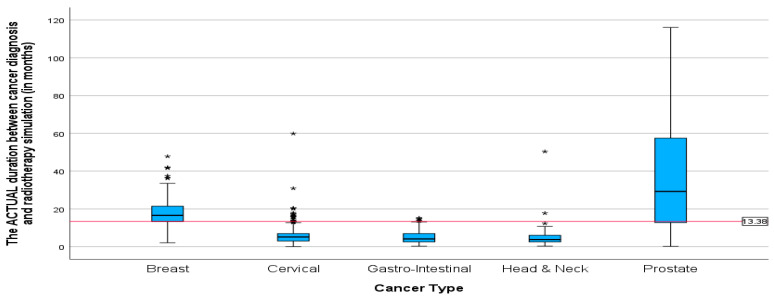
Boxplots showing duration between cancer diagnosis and CT simulation for different cancers in months. * Indicates the cases in the actual duration.

**Figure 8 ijerph-22-00544-f008:**
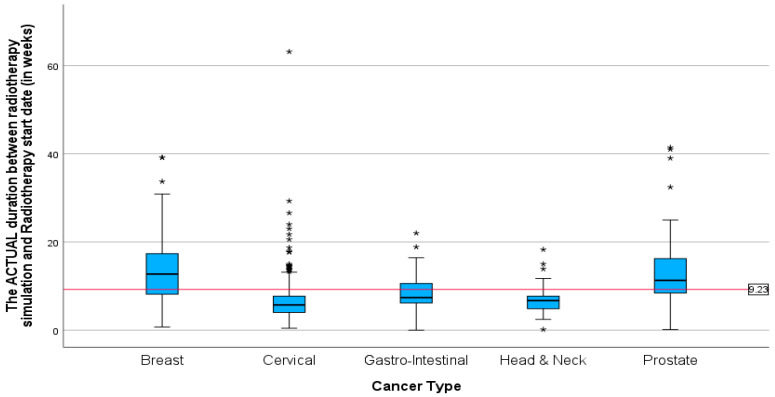
Boxplots showing the duration between CT simulation and the start of radiotherapy for different cancers in weeks. * Indicates the cases in the actual duration.

**Table 1 ijerph-22-00544-t001:** Sample Characteristics.

Characteristic	Value/Category		N	Percentage (%)
Gender	Female		571	71.4%
	Male		229	28.6
Age	19–25		2	0.3
	26–35		37	4.6
	36–45		172	21.5
	46–55		230	28.8
	56–65		198	24.8
	66–70		87	10.9
	>70		74	9.3
Cancer Type	Breast		146	18.3
	Cervical		372	46.5
	Gastro-Intestinal		76	9.5
	Head and Neck		71	8.9
	Prostate		135	16.9
Stage at Diagnosis	Stage 1		50	6.3
	Stage 2		287	35.9
	Stage 3		383	47.9
	Stage 4		80	10.0
Referral pattern	Johannesburg	CHBAH	164	20.5
		Internal	76	9.5
		Other Provinces	0	0
	Pretoria	DGMH	110	13.8
		SBAH	106	13.3
		Other Provinces	98	12.3

**Table 2 ijerph-22-00544-t002:** Distribution of diagnosis and stages.

Facility	Stage 3 and 4 N (%)	Stage 3 and 4 N (%)
CMJAH	178 (44.5%)	222 (55.5%)
SBAH	159 (39.8%)	241 (60.3%)

**Table 3 ijerph-22-00544-t003:** Key time intervals Statistics.

Activity	Mean (SD ^1^)	Median	Min–Max	95% Confidence Interval of the Difference Lower–Upper
D1	8.329 (10.906)	4.961	−39.33–114.33	2.637–5.867
D2	13.382 (17.249)	6.883	0–116.04	7.576–12.632
D3	9.229 (6.333)	7.571	0–63.14	1.517–3.343

D1—the actual duration between diagnosis (biopsy date) and the first consultation with a radiation oncologist (months); D2—the actual duration between cancer diagnosis and CT simulation for radiation planning (months); D3—the actual duration between CT simulation for radiotherapy planning and radiotherapy start date (weeks). ^1^ SD = standard deviation, Min = minimum, Max = maximum.

**Table 4 ijerph-22-00544-t004:** Participants’ consultation timeframe with a radiation oncologist.

Activity	Mean (SD)	Median	Min–Max
Hospital presentation (biopsy)	8.329(10.906)	4.961 months	39.33–1114.33

SD = standard deviation, Min = minimum, Max = maximum.

## Data Availability

All data in this study were provided in the main manuscript.

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
