# Peer review of "Improving Access to Radiotherapy Services in Gauteng: Quantitative Analysis of Key Time Intervals from Diagnosis to Treatment"

_ijerph, 2025, doi:10.3390/ijerph22040544_

Round 1
Reviewer 1 Report
Comments and Suggestions for Authors
Thank you for inviting me to review this manuscript.
As it stands, the manuscript is clearly written and well presented but (in my view) simply represents an analysis of patient access to radiotherapy services in two centers for a period of one year - without an analysis of why differences existed and therefore cannot claim that it contributes to addressing how to improve 'access'.
This means some aspects of the manuscript are misleading.
For example, the title states 'Improving Access to Radiotherapy Services in Gauteng: '. But, the study focuses on (i) when patients got access to various radiotherapy services and (ii) differences in patient acess to treatment in two centers: Centre in Johannesburg and the Centre in Pretoria. It does not focus on 'improving' access.
The study analyses one year of patient access data to both centers. With the exception of lines 266-268, it does not explore why access differences exist between both centers.
If the study did attempt to investigate "delays in referral, scheduling issues, limited access to specialists, or patient-related factors", then that data analysis could inform readers on factors that affect patient access to radiotherapy services and in turn, that could be used to inform a discussion on Improving Access to Radiotherapy Services in Gauteng.
As it stands, the content of this manuscript does not match it's title.
Additionally, it does not explore the factors that affect access to radiotherapy services. The research team may be in a position to do so by investigating referral, scheduling issues, access to specialists, or patient-related factors", in both centers.
Again, many thanks for forwarding the manuscript.
Reviewer 2 Report
Comments and Suggestions for Authors
Dear authors,
Thank you for your contribution and submission.
This is an interesting paper. It is timely and addresses an important topic that aligns well with sustainable development goal number three related to health and well-being. We need more studies like this as we believe such studies will help countries on the road towards universal health coverage.
Overall, it is well-written and informative. The research design is well-planned and well-implemented. I have few review comments, suggestions, and recommendations for your consideration as follows.
Abstract
line 24: please add what CT abbreviation stands for as this is the first time it appears in text.
keywords: it is recommended to arrange keywords in an alphabetical order.
Materials and Methods
Study Participants: can you be more specific in terms of inclusion and exclusion criteria? also, please add recruitment method.
Results
Sample Characteristics: for better readability and visual representation of demographics, may I suggest putting all data into a table?
Discussion
Strengths and limitations
line 307: "..however, the qualitative aspect will be explored in another phase of the study". Would you please elaborate? readers would be interested in knowing more about future directions for future research.
I will be happy to look at the revised version.
Many thanks.
Best wishes,

Reviewer 3 Report
Comments and Suggestions for Authors
- Main Question and Explicitness:
While not stated as a single, concise question in the introduction, the objective was clearly implied throughout the abstract and introduction. The study "aims to quantitatively analyze the key time intervals..." (lines 16-18) is the most direct statement of the core research question. The introduction established the context of limited access to radiotherapy, particularly in low- and middle-income countries (LMICs) like South Africa, and frames timely access as critical for patient outcomes.
- Originality, Relevance, and Gap Addressed:
- Originality and Relevance: The study's originality lies in its specific focus on the Gauteng province of South Africa, providing localised data that are often lacking in LMIC contexts. While numerous studies have examined radiotherapy access globally, this granular, regional analysis is crucial for informing targeted interventions. The study is highly relevant to the field of public health, radiation oncology, and health systems research in resource-constrained settings. The findings about referrals and time lines between the different treatment sites provided more critical practical information than other similar health care systems.
- Gap Addressed: The paper directly addresses the gap in understanding the specific time intervals and bottlenecks within the radiotherapy pathway in Gauteng. While prior research has documented access challenges (cited appropriately, e.g., [2-4], [6]), this study provided quantitative data on where the delays occurred (diagnosis to consultation, consultation to CT simulation, simulation to treatment). This moved beyond general statements about limited access and provided actionable insights for system improvement. This moved past broad acknowledgment of access challenges toward quantification and identification.
- Methodological Improvements and Controls:
The methodology was generally sound, employing a retrospective, cross-sectional design. However, several improvements could be considered:
- Sampling Methodology: While the study mentioned a "systematic sampling method" (line 96), more detail is needed. How were the "regular intervals" defined? Was it every nth file, or was there a random element within the interval? Providing the specific interval (e.g., every 3rd file) would enhance reproducibility.
- Sample Size Justification: Although 800 patient records were a substantial sample, a power analysis to justify this number is absent. The authors should explain why 800 was deemed sufficient to detect meaningful differences in time intervals. Was it based on prior studies, expected effect sizes, or practical limitations? Adding a brief justification would strengthen the methodology.
- Missing Data: The manuscript did not explicitly address how missing data in patient files were handled. Were files with incomplete time interval data excluded? Were there any imputation methods used? Transparency about handling missing data is crucial for assessing the robustness of the findings. (This should be added around line 93).
- Confounding Factors (Lines 40-41, and generally): While the study identified delays, it did not fully explore why these delays existed. Factors, such as socioeconomic status, distance to the treatment center, specific cancer stage (beyond a basic description), and insurance status (if relevant in South Africa) could influence waiting times. While a full causal analysis might be beyond the scope, acknowledging these potential confounders and, if possible, collecting data on a few key ones (even basic demographics) would greatly enhance the study. A multiple regression analysis could then be used to adjust for these confounders.
- Study Dates: Line 20 describes data collection from January to December 2023, whereas the copyright year and dates in the reference list are set as 2024. Please resolve these discrepancies.
- Retrospective Nature: The authors should more explicitly acknowledge the limitations inherent in a retrospective study. For example, reliance on existing records means the researchers had no control over the quality or completeness of the original data collection. (This limitation should be mentioned explicitly in the Discussion/Limitations section, not just briefly implied in the Strengths and Limitations section).
- Conclusions, Evidence, and Addressing Questions:
- Consistency with Evidence: The conclusions were largely consistent with the evidence presented. The authors found significant delays in accessing radiotherapy and identify specific points in the pathway where these delays were most pronounced. The statistical analysis (ANOVA, t-tests, Chi-squared) supports the conclusions drawn about differences between centers and cancer types.
- Addressing Main Questions: The study did address its main implicit question by quantifying the key time intervals. It clearly showed the duration of each interval and highlighted the variability between centers and cancer types. However, as mentioned above, it fell short of fully explaining the causes of these delays. It described what was happening, but less so why.
- Oversimplification: The conclusion (Section 6) could be strengthened by being more specific and action-oriented. Phrases, such as "a multifaceted approach is required" (line 319) are vague. The authors should draw more directly on their findings to suggest concrete, prioritised interventions. For example, "Based on the finding that the diagnosis-to-consultation interval is particularly long, we recommend prioritising interventions to streamline the referral process and reduce specialist appointment wait times."
- Tables, Figures, and Data Quality:
- Tables: The tables (Table 3 and Table A1) are generally clear and well-organised. They effectively present the descriptive statistics for the key time intervals. However, consider adding confidence intervals to the mean values to provide a measure of precision.
- Figures: The figures (histograms and boxplots) are informative and visually appealing. They effectively illustrate the distribution of waiting times and highlight differences between groups. However:
- Figure 1: The y-axis label "Count" is redundant since it is already a histogram. Just labeling the y-axis is sufficient. More importantly the bars in the histograms should be differentiated by colours or similar design in terms of showing different cancers/stages. It is very difficult to see from this design what part of each bar represents which state.
- Figures 2-7: These histograms need units. The 'count' of patients needs to be shown clearly. All y-axes for graphs need clarification of whether values are given in means or other metrics. The 'dots' represented outliers; it would be great to have the meaning of the shapes explicitly defined within the image to avoid the reader needing to search the text.
- Figures 8-10: Provide p-values directly on the figures (e.g., above the boxplots) to show which comparisons are statistically significant. This enhances visual interpretation.
- Data Quality: The manuscript did not delve into the quality control measures taken during data extraction. Were two independent reviewers used to minimise extraction errors? Was there a process for resolving discrepancies? Describing these steps would enhance confidence in the data quality. (This would fit well in the Data Collection section).
- Caveats, Weaknesses, and Mistakes:
- Line 19: "From the two public radiotherapy centres" - It would be helpful to name the centers (if ethically permissible and approved) or at least describe their characteristics (e.g., "one urban tertiary care center and one regional hospital").
- Line 45: "...at significant personal cost, to access treatment." - This statement, while likely true, needs either a citation or a brief elaboration. What were the specific costs (transportation, lost wages, accommodation)?
- Lines 49-51, referring to "12-13": While true that providers may lack info, these specific lines refer more to patient challenges, especially line 49, which refers more to population access than professional provider information. A better in text citation may be useful.
- Line 54: "...enhancing quality of life, and ensuring equitable access to life-saving cancer care [14-16]." The references provided, particularly 14, do not fit well with the entire sentence - ref. 14, in particular is specific for breast cancer radiotherapy.
- Line 96: "...a systematic sampling method". Needs greater definition of what a "systematic sampling method" means in the context of this work.
- Line 106: "Andersen Newman's behavioural model..." This model should be briefly described, before it is used to define key terms. The reader needs a basic understanding of the model to understand how "access" is being conceptualised. Add a sentence or two explaining the model's core concepts before using it.
- Line 140, regarding table A1 and figure 1: These data presentations can be confusing. It will help the flow and reader to present figure 1 much earlier (e.g. just after the reference to Table A1). As presented in the submission, figures are found many pages away from their text references.
- Line 193: "The minimum value of -39.33 months is an anomaly..." This needs further investigation. It suggests a patient presented at radiation oncology before diagnosis. While possible (perhaps a very rapid referral for suspected cancer), it needs to be explained. Was it a data entry error? Was there a clinical reason for this unusual sequence? The explanation "for some unstated reason" is insufficient. The authors need to go back to the original record (if possible) and clarify this.
- Lines 325-327: The contributions described appear repetitive ("writing...writing..."). Refine this to more clearly distinguish the roles. Also, consider adding a statement about who had access to the full dataset.
- Throughout: There are minor grammatical and stylistic issues (e.g., awkward phrasing, occasional tense inconsistencies). A thorough proofread by a native English speaker would be beneficial.
Round 2
Reviewer 1 Report
Comments and Suggestions for Authors
Thank you for sending me the revised version of your manuscript.
This version provides a much more comprehensive account of the overall research and appropriately situates this manuscript in that work.
Reviewer 3 Report
Comments and Suggestions for Authors
Glad with changes